

# Design of a 3D emotion mapping model for visual feature analysis using improved Gaussian mixture models

Enshi Wang[1] and Fakhri Alam Khan[2,3,4]

[1] School of Digital Art, Wuxi Vocational College of Science and Technology, Wuxi, Jiangsu, China
[2] Information and Computer Science Department, King Fahad University of Petroleum and Minerals, Dhahran, Saudi Arabia
[3] SDAIA-KFUPM Joint Research Center for Artificial Intelligence, KFUPM, Dhahran, Saudi Arabia
[4] Interdisciplinary Research Center of Intelligent Secure Systems (IRC-ISS), KFUPM, Dhahran, Saudi Arabia

## ABSTRACT

Given the integration of color emotion space information from multiple feature sources in multimodal recognition systems, effectively fusing this information presents a significant challenge. This article proposes a three-dimensional (3D) color-emotion space visual feature extraction model for multimodal data integration based on an improved Gaussian mixture model to address these issues. Unlike traditional methods, which often struggle with redundant information and high model complexity, our approach optimizes feature fusion by employing entropy and visual feature sequences. By integrating machine vision with six activation functions and utilizing multiple aesthetic features, the proposed method exhibits strong performance in a high emotion mapping accuracy (EMA) of 92.4%, emotion recognition precision (ERP) of 88.35%, and an emotion recognition F1 score (ERFS) of 96.22%. These improvements over traditional approaches highlight the model's effectiveness in reducing complexity while enhancing emotional recognition accuracy, positioning it as a more efficient solution for visual emotion analysis in multimedia applications. The findings indicate that the model significantly enhances emotional recognition accuracy.

## INTRODUCTION

In today's digital age, the rapid advancement of multimedia technology has enabled widespread access to and sharing vast amounts of visual and auditory content. In this era of information explosion, it is increasingly crucial to understand and convey the emotional information embedded in these multimedia contents (*Chung & Cheon, 2020*). Visual aesthetic features, including color, composition, texture, and other aspects, are pivotal in expressing emotions within multimedia content (*Wu, Yan & Huang, 2023*).

Machine learning techniques, such as support vector machines, neural networks, and multiclass classifiers, are frequently employed in image recognition and aesthetic attribute categorization (*Pan et al., 2024*; *Jin, Wang & Meng, 2024*; *Shi, Hayat & Cai, 2024*).

Corresponding author
Enshi Wang, 2056016@wxsc.edu.cn

Additionally, evolutionary algorithm techniques and regression models have been utilized to predict the aesthetic qualities of images based on low-level statistical texture features (*Groissboeck, Lughofer & Thumfart, 2010*). While machine learning models offer a flexible general modeling approach, they are often considered opaque and uninterpretable. To enhance interpretability, several neuropsychological and neuroaesthetic models for visual texture aesthetics have been proposed (*Li & Wang, 2024*). A neuropsychological model grounded in the neurological principles of visual information processing was introduced for visual texture aesthetics.

Furthermore, scholars have developed a significant framework for the experience and evaluation of aesthetics, detailing the information flow across various circuits within the brain's neural networks (*Yoo, Jasko & Winkielman, 2024*). According to *Koelsch et al. (2015)*, in the comprehensive neurofunctional model of human emotions, four fundamental brain circuits are involved in emotion processing. Redies presents a unified approach to visual aesthetic experience by integrating cultural context with enduring notions of beauty (*Redies, 2015*). *Chen et al. (2024)* developed a hierarchical feedforward model to explore the relationship between computational texture attributes and visual texture aesthetic qualities, informed by psychological research. Subsequent studies have further expanded on this work (*Hu et al., 2024*).

*Wang et al. (2023)* and *Zhu (2023)* reviewed 75 methods across four categories—data source types, affective computing, visual coding and visualization, and visual analysis tasks—along with 15 subcategories covering visual works mentioned in published articles and interactive visual works available online. Despite these advancements, emotion recognition accuracy in multimedia applications remains challenging, particularly with high-dimensional and complex data. Moreover, data processing often generates significant, redundant information, complicating model complexity. With its robust learning capabilities, deep learning can extract advanced semantic features and address gaps in data representation (*Gandhi et al., 2023*). In this context, this study aims to design a three-dimensional color emotion space mapping model based on visual aesthetic features. This model integrates machine vision technology to extract rich information from visual content and to model and express emotional states. In developing this emotion model, we considered emotion layer perception, judgment layer perception, and emotion layer integration to create a more comprehensive emotional representation.

This study aims to experimentally validate the performance of the proposed model and compare it with other methods. We will assess the model's performance across various contexts by introducing several experimental indicators, such as emotion mapping and recognition accuracy. Additionally, we will examine the model's robustness by testing its performance under varying interference levels. Finally, we will analyze the experimental results to provide insights for further model refinement.

The main contributions of this study are as follows:

Integration of visual aesthetic features: This study introduces a method based on visual fusion and the Gaussian Mixture Model, which integrates visual aesthetic features, including color, composition, texture, clarity, depth, emotional content, and dynamic elements. This comprehensive approach aims to provide a more detailed understanding of

emotional states in multimedia content, offering a holistic perspective for emotional modeling.

Multimodal data integration: The proposed method combines visual and auditory elements within multimedia content to form a more complete emotional model. Notably, vision and hearing are not independent but interconnected, enhancing the depth of emotional representation.

## RELATED WORK

In digital media, the proportion of individuals obtaining information through visual perception rapidly increases. The widespread dissemination of images and videos facilitates the sharing and acquisition of vast information, yet conveying emotions through these media remains a complex challenge (*Liu, Lughofer & Zeng, 2017*). Traditional sentiment analysis methods are often too simplistic to accurately capture the nuanced emotions in multimedia content (*Muratbekova & Shamoi, 2024*). Recent advancements in cognitive neuroscience, neurobiology, and mathematical sciences have led to various visual beauty models to explore how different visual textures evoke specific emotions and experiences (*Wang et al., 2023*). Current research in sentiment analysis predominantly focuses on three modalities: text, image, and audio, with a particular emphasis on text sentiment analysis. Significant efforts by scholars both domestically and internationally have greatly improved the accuracy of text sentiment classification. However, as image, text, and video data become increasingly prevalent, single-modal sentiment analysis faces limitations such as the "emotional gap" and "ambiguous irony," which hinder the effective identification of emotions in such information. Consequently, there is a growing shift towards multimodal approaches.

Emotional analysis seeks to enable computers to extract the emotional nuances from visual information, with color being a fundamental element of visual art that significantly influences art perception (*Wędołowska, Weber & Kostek, 2023*). The effectiveness of sentiment analysis models heavily depends on the quality of sentiment dictionaries, making this research area crucial within natural language processing (NLP). Currently, sentiment dictionaries primarily contain word sentiment information. To address the limitation of simultaneously assessing sentiment and sentiment intensity, *Abdykerimova et al. (2024)* proposed an approach for automatically creating a fine-grained sentiment dictionary that includes both sentiment and intensity information. *Ananthram, Winn & Muresan (2023)* aims to use machine–learning methods to predict emotions based on the colors in images and video excerpts. Unlike traditional sentiment analysis, machine learning-based text sentiment analysis does not rely on sentiment dictionaries. Instead, it employs traditional machine learning techniques, such as support vector machines (SVM) and random forests, to model data through annotated training and predict results (*Das, Biswas & Bandyopadhyay, 2023*). Despite extensive research into the relationship between color and emotion, the impact of context-based color changes on the perceived intensity of emotions remains underexplored. A novel multimodal dataset introduced by *Chen et al. (2024)* investigates how lines, strokes, textures, forms, and language influence the emotional meanings associated with color.

*Liu et al. (2024)* utilized convolutional neural networks to extract image features and employed multi-layer perceptrons for sentiment classification, achieving superior performance compared to traditional models. However, conventional convolutional neural networks, such as VGG Net and AlexNet, are typically designed for entity classification within the center of the image. These networks focus on extracting semantic relationships and entities through continuous convolution and pooling, but they often fall short in extracting low-level visual and mid-level aesthetic features. To address this, *Zeng et al. (2024)* proposed a multi-level image depth representation method incorporating high-level, mid-level, and low-level visual features, enhancing sentiment classification accuracy across various image types.

Sentiment analysis encounters new challenges when addressing multimodal sentiment analysis (MSA) with unclear or missing modalities. An efficient MSA model that considers missing patterns has been developed to address this issue. However, existing approaches often overlook the complex relationships among multiple modalities and primarily utilize serial processes for feature fusion. *Ananthram, Winn & Muresan (2023)* proposed a Modal Translation-based MSA model (MTMSA) to handle unclear missing modalities. *Zhu et al. (2023)* introduced a text-centered hierarchical fusion network (TCHFN) that combines text, video, and audio modalities using a hierarchical fusion strategy. In this framework, low-level fusion involves cross-modal interactions between paired modalities, while high-level fusion extends these interactions to encompass all three modalities. *Devillers, Maytié & VanRullen (2024)* proposed a novel multimodal dataset for investigating the emotional connotations of colors as mediated by lines, strokes, textures, shapes, and language.

Recent research has predominantly focused on non-temporal fusion techniques. *Qian et al. (2024)* introduced a tensor fusion method that employs the Cartesian product to combine single-modal features extracted by a temporal network. After extracting features from different modalities, techniques such as distribution similarity, orthogonality loss, reconstruction loss, and task prediction loss are used to learn modality-invariant and modality-specific representations. Innovative methods such as graph capsule networks have been proposed recently and have demonstrated promising results.

## METHODOLOGY

To address the limitations of traditional visual emotion modeling methods, we propose an innovative three-dimensional (3D) color-emotion visual feature extraction model designed explicitly for multimodal data integration. Our approach harnesses the capabilities of machine vision technology and advanced statistical modeling to capture and express emotional states within multimedia content accurately. The core components of our method include data collection and preprocessing, feature extraction using an enhanced Gaussian mixture model (GMM), feature fusion, and the development of a multi-layer emotion perception model. Our approach involves several key steps: (1) keyframe extraction based on machine vision, (2) working sequence potential function establishment, (3) visual feature extraction using an improved Gaussian mixture model, and (4) feature fusion using entropy and visual sequences. These steps are designed to handle complex multimodal data and enhance emotion recognition accuracy.

## Keyframe extraction based on machine vision

The process begins with extracting keyframes from multimedia content to identify representative frames. This is achieved by performing optical flow calculations, where each frame's entropy value is computed. Frames with the highest entropy values are selected, ensuring that the most informative content is retained for further analysis. Keyframe extraction is selecting the most representative and informative frames in a multimedia sequence to reduce storage and processing costs while preserving important multimedia content information. Firstly, we perform optical flow calculations on the relevant multimedia content sequences, represented as follows:

$$E(w) = \beta E_{color}(w) + \gamma E_{grad}(w) + \alpha E_{smoorh}(w) \tag{1}$$

where $E_{color}(w)$ represents the assumption of brightness invariance; $\alpha$, $\beta$ and $\gamma$ represent adjustable weight parameters, respectively. By introducing gradient constraint $E_{grad}(w)$ to reduce the impact of lighting, and using $E_{smoorh}(w)$ to smooth the video. Calculate the entropy value corresponding to the current optical flow chart using Formula (2).

$$E\_img = -\sum_{k}^{m} \log_2 p_k E(w) \tag{2}$$

where $E\_img$ represents entropy value; $m$ represents grayscale level; $p_k$ represents the proportion of pixels with a grayscale value of $k$ in the image. The larger the entropy value, the more information in the image.

As the primary step for extracting multimedia energy features in multimedia signal processing, we initiate the process by windowing and framing the multimedia signal $x(j)$, obtaining the $k^{th}$ frame multimedia. The multimedia signal is stored in an array $y$ of length $N$, with each window sampled at a length of $wlen$ and a sampling frequency of $fs$. $Olap=wlen-dis$ describes the overlapping part between two frames, while $dis$ represents the displacement between consecutive frames. This framing process is based on Eq. (3) and is applied to the multimedia signal of length $N$. It lays the foundation for subsequent feature extraction and the mapping of emotional space in our design based on visual aesthetic features:

$$fs = \frac{N - olap}{dis} = \frac{N - wlen}{dis + 1}. \tag{3}$$

Then, by applying Eq. (4), the average amplitude of the video is calculated to obtain the energy feature corresponding to the video:

$$\begin{cases} y_k(j) = win(j) \times x[dis(k - 1) + j]E\_img \\ M(k) = \sum_{j=0}^{L-1} |y_k(j)|fs \end{cases} \tag{4}$$

where $y_k(j)$ represents the values of a single frame, $win(j)$ denotes the window function, and $M(k)$ signifies the energy magnitude corresponding to a video frame.

The process mentioned above achieves the alignment of entropy sequences and feature sequences. Subsequently, a multiplication operation is performed between the entropy sequence and the visual feature sequence to accomplish feature fusion, acquiring entropy sequences associated with the visual aspect.

Calculate the entropy sequence corresponding to the fused features. Using the following formula, frames with values exceeding the threshold are selected as key frames for the visual aspect by comparing the obtained values with a threshold. The expression for extracting key frames is as follows:

$$V = \frac{|H_{current} - H_{key}|M(k)}{H_{ky}} \tag{5}$$

where $H_{key}$ and $H_{current}$ are the frame values corresponding to key frames and the entropy values corresponding to the current frame, respectively.

## Feature extraction

### Feature recognition

Based on the obtained keyframes, visual feature extraction is conducted in two steps: background elimination and feature extraction. Background elimination is achieved using a GMM, and the process is outlined as follows:

Establish the model, where $X_t$ represents the corresponding value of a certain pixel at time $t$, and $P(X_t)$ represents the probability of $X_t$ occurring. The expression is as follows.

$$P(X_t) = \sum_{i=1}^{K} \omega_{i,t} \times \eta(X_t, \mu_{i,t}, \sigma_{i,t}) V \tag{6}$$

where $\omega_{i,t}$ represents the weight of the $i$-th Gaussian distribution at time $t$, $\sigma_{i,t}$ represents the variance, $\mu_{i,t}$ represents the mean, and $\eta(X_t, \mu_{i,t}, \sigma_{i,t})$ represents the probability density function. The calculation method for the probability density function $\eta(X_t, \mu_{i,t}, \sigma_{i,t})$ is as follows:

$$\eta(X_t, \mu_{i,t}, \sigma_{i,t}) = \frac{P(X_t)e^{-\frac{1}{2}(X_t - \mu_{i,t})^T \sigma_{i,t}^{-1}(X_i - \mu_{i,t})}}{\sqrt{2\pi|\sigma_{i,t}|}} \tag{7}$$

Assuming the value of a pixel in a frame image is $X_t$, use $|X_t - \mu_{i,t}|$, $2.5\sigma_{i,j-1}$ to determine whether $K$ Gaussian distributions can match with this pixel. Update the weight, variance, and mean of the Gaussian distribution through Eq. (8):

$$\begin{cases} \omega_{i,t} = (1 - \alpha)\omega_{i,t} + \alpha \\ \mu_{i,t} = (1 - \beta)\mu_{i,t-1} + \beta X_{i,t} \\ \sigma_{i,t} = (1 - \beta)\sigma_{i,t-1} + \beta(X_{i,t} - \mu_{i,t})^T(X_{i,t} - \mu_{i,t}) \\ \beta = \alpha\eta(X_i, \mu_{i,t}, \sigma_{i,t}) \end{cases} \tag{8}$$

where $\alpha$ represents the Gaussian mixture model taking values within the interval $[0, 1]$,

which the update speed of the image background model is controlled by the parameter $\alpha$, and $\beta$ represents the parameter update factor determining the speed of parameter updates.

Foreground detection. After completing the training of the background model, arrange the $K$ Gaussian distributions based on the size of $\lambda_{i,t}$, selecting the top $B$ Gaussian distributions to form the background.

$$B = \arg\min\left(\sum_{k=1}^{b} \omega_k > T\right). \tag{9}$$

To provide a basis for the sequence potential function of relevant visual tasks, we use a threshold recognition algorithm to identify visual tasks. The specific process is as follows:

1) Let $N$ represent the number of pixels in the image. Describe the motion image using matrices and utilize $A(x_1, y_1)$, $B(x_2, y_2)$, $C(x_3, y_3)$, and $D(x_4, y_4)$ to denote the coordinates of points. Calculate the parameters $P$ and $S$ through Eq. (10).

$$\begin{cases} P = \dfrac{x_2 - x_1}{y_3 - y_1} \\ S = (x_2 - x_1) \times (y_3 - y_1) \end{cases}. \tag{10}$$

2) During the motion recognition process, if the pixel value exceeds the threshold A, it is considered N.

3) If no pixels are within the specified range, update the coordinates and expand the search area in the aerobics exercise motion image.

4) Complete the scanning process and obtain $N_e$ and recognize the target.

5) Calculate the aspect ratio of the target region and compare the result with the recognition threshold. If the threshold $A$ exceeds the aspect ratio $|1 - P|$, proceed to the next step.

6) Calculate the size of the target region, obtain the ratio $M = N_e/S$ of the target area to the motion image area, and compare this ratio with the size of the threshold $A$. When $|0.785 - M| < A$, it signifies the recognition of the motion.

## Working sequence potential function

Once the keyframes are extracted, we establish a working sequence potential function using the Harris3D operator. This function identifies and evaluates motion-related visual features in the video, facilitating the recognition of specific actions or emotions. The function calculates the shortest Euclidean distance between local reference points and spatiotemporal points in the video, allowing the system to detect relevant motion sequences efficiently.

Utilizing the Harris3D operator based on motion recognition results to establish the sequence potential function, laying the foundation for visual feature extraction.

Let $(X_{zi}, Y_{zi})$ represent the key points in the video, with the local reference point as $(a_i, b_i)$ and $n$ denotes the shortest Euclidean distance between the local reference center point and the spatiotemporal point, and its calculation formula is as follows:

$$n = \frac{\arg\min \sqrt{(a_i - x_j)^2 + (b_i - y_j)}}{(x_{zi}, y_{zi})} \tag{11}$$

where $(x_i, y_i)$ represents the spatiotemporal point.

Obtain the action dataset through k-means clustering, where $f_p$ represents the BOW feature corresponding to the action image $p$. The calculation method is:

$$f_p = \frac{K_n \times N}{K} \times \frac{K_n \times 162}{p} \tag{12}$$

where N represents the spatiotemporal unit sequence length corresponding to the action image, and Kn represents the number of cluster centers within the n range.

Fuse the BOW features through the following expression:

$$F_p = \sum_{n \in [1,7]} K_n \times N f_p \tag{13}$$

where $F_p$ represents the fused features within various levels of the action image.

Now, based on the above analysis, we can obtain the working conditional probability model as follows:

$$P(Y, h/X, \theta) = \{X_i\}_{i=1}^t F_p \times \frac{\exp(f_p \cdot \varphi(Y, h, Z))}{\sum (F_p \cdot \varphi(Y, h, X))} \tag{14}$$

where $\varphi(Y, h, X)$, $Y$, $\theta$, $h$, and $\{X_i\}_{i=1}^t$ represent the sequence potential function of actions, the sequence label, represents a constant, represents the hierarchy, and any action sequence, respectively.

For different levels of the working sequence potential function, the calculation is performed as follows:

$$\varphi(Y, h, X) = \left[\sum \varphi_1(X_j, h_j) \sum \varphi_2(Y, h_j)\right] \frac{\sum \varphi_3(Y, h_j, h_k)}{P(Y, h/X, \theta)} \tag{15}$$

where $\varphi_1(X_j, h_j)$ represents the relationship between the prediction node and the latent variable node and $\varphi_2(Y, h_j)$ represents the relationship between the sequence label point and the latent variable node.

Let $(x_1, y_1)$, $(x_2, y_2)$, and $(x_N, y_N)$ represent the sample dataset, where $x_i$ denotes the sample data, and $y_i$ represents the sample labels. Utilizing the AdaBoost algorithm based on calculating the sample error rate.

$$\varepsilon_t = \frac{\varphi(Y, h, X) \times (h_t(x_i) \neq y_i)}{(x_1, y_1), \cdots, (x_i, y_i), \cdots, (x_N, y_N)}. \tag{16}$$

Based on the above results, extract visual features to obtain:

$$\hat{C}_i = \arg\min ||d_i - C(d_j)^2|| \times \varepsilon_t \qquad (17)$$

where $d_i$ represents the $j$-th feature data present in the aerobics jumping action sequence, $C(d_j)$ represents the category to which the $j$-th aerobics jumping feature data belongs in the training sample. Utilizing the model constructed by the above formula, complete the extraction of visual features.

## Three-dimensional color-emotion space model

Before establishing the model, it is essential to smooth and normalize the feature data to ensure the model's robustness. This process involves normalizing the data using the maximum-minimum scaling method.

$$r' = \frac{r - \min(X)}{\max(X) - \min(X)} \qquad (18)$$

where the least and maximum ratings in dataset X are denoted, respectively, by min(X) and max(X).

Building models for visual texture aesthetics represents a distinctive application of machine learning. These models aim to bridge the gap between high-level aesthetic perceptions and low-level texture properties. In this section, we utilize six activation functions to construct the emotional model, comprising the emotion, judgment, and perception layers.

The primary function of the emotion layer is to extract emotional information from the input visual aesthetic features. Within the three-dimensional color-emotion space mapping model, the emotional layer perception model is an intermediary between high-level emotional perception and low-level texture features. Its construction is detailed as follows:

$$G = F_1(M) + R_0 \qquad (19)$$

The primary function of the judgment layer is to make assessments or evaluations, thereby identifying or quantifying the position or attributes of input visual aesthetic features in the emotional space. The judgment layer perception model plays a role in categorizing, scoring, or positioning visual features, contributing to the understanding and expressing emotional information associated with these features. Its construction involves:

$$T = F_2(M) + F_3(G) + R_1 \qquad (20)$$

The primary function of the perception layer is to capture and represent emotional information related to visual aesthetic features in the emotional space. It focuses on extracting emotional features from the input and understanding the distribution and relationships of these features in the emotional space through model construction. The specific construction involves:

$$Q = F_4(M) + F_5(G) + F_6(T) + R_2 \qquad (21)$$

where $F_i(i = 1, 2, 3, 4, 5, 6)$ represent six linear/non-linear activation functions, $R_0$, $R_1$, and $R_2$ represent emotional thresholds, *i.e.*, the minimum values of aesthetic attribute scores; the symbol + indicates the accumulation of emotions through different perceptual stages.

This section employs a comparative method to calculate different sets of emotions and construct an emotional space model. After representing emotions in a three-dimensional space, we calculate the average angle of each emotion to its origin. This angle defines the shortest rotation angle for any emotion around the origin and is responsible for assuming the establishment of emotions.

To compute the emotional space in three dimensions, we calculate the average angle for each emotion to construct an emotion wheel and obtain definitive patterns. To maintain normality, we also exclude some outliers and data points unusually distant from other values in the dataset. The standard formula defines the angle between two vectors.

$$\cos \Theta (U, V) = \frac{Dot\ Product}{Eucledian\ Norm}$$
$$\cos \Theta(U, V) = \frac{(U, V)_R}{||U||\ ||V||} \tag{22}$$
$$\Theta = ArcCos(\cos(U, V))$$

where $(U, V)_R$ defines scalar product for any pairs of vectors $U, V \in X_R$ in any real vector space $X_R$. The scalar product is defined as:

$$(U, V)_R = \sum_{i=1}^{n} U_i V_i; \text{where } n \geq 2 \tag{23}$$

And

$$||U|| = \sqrt{(U, U)_R}; ||V|| = \sqrt{(V, V)_R} \tag{24}$$

where $|U||$ and $||V||$ denote the Euclidian norm or magnitude of respective vectors U and V.

Based on the above analysis, the flowchart of the proposed method in this article is shown in Fig. 1.

The critical steps in our proposed method are as follows:

1. Data collection and preprocessing:

A diverse dataset comprising various images annotated with emotional labels was collected. This dataset included images from publicly available sources and specifically curated data for our experiments.

2. Feature extraction using improved GMM:

We enhanced the GMM to handle high-dimensional visual data more effectively. This involved fitting a mixture of multiple Gaussian distributions to the feature space, capturing the underlying structure and complexity of the data.

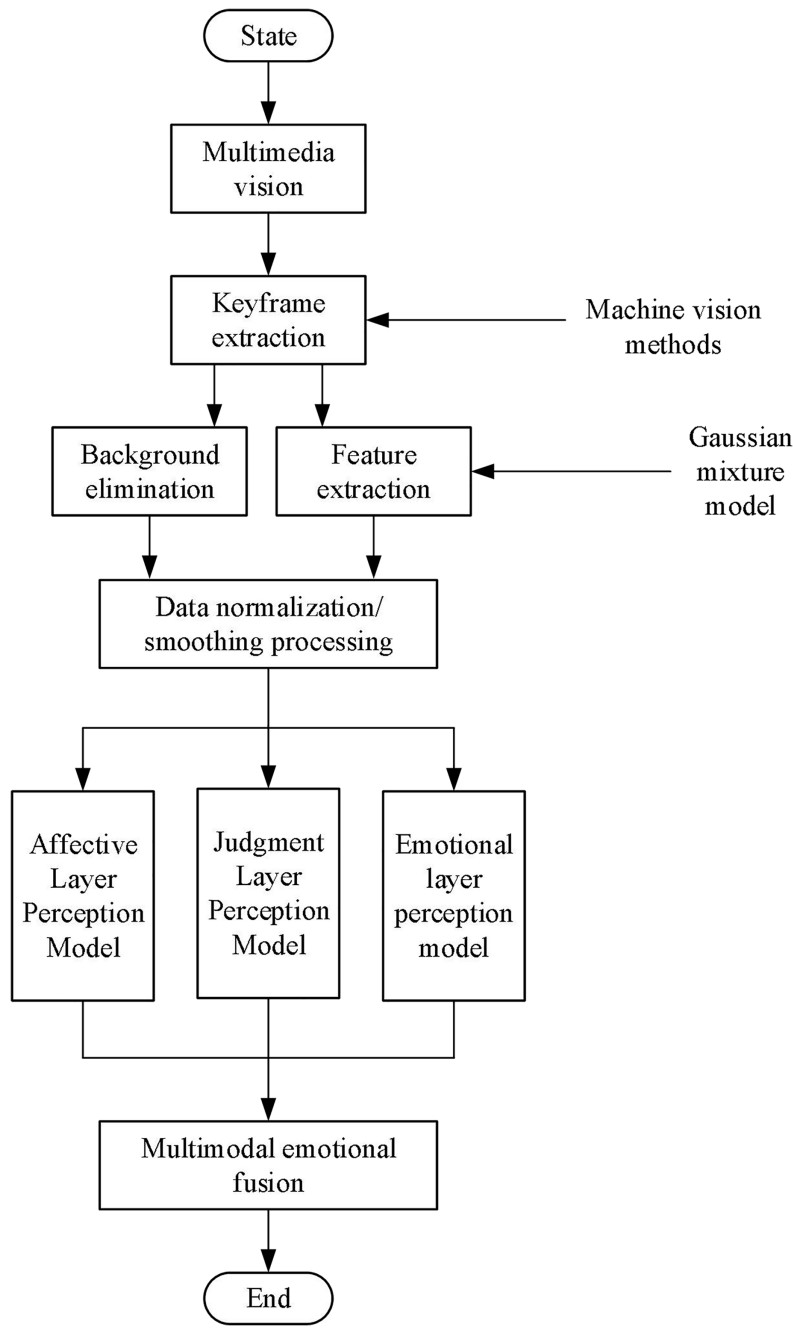

**Figure 1  Block diagram of the proposed method.**

3. Feature fusion using entropy sequences and visual feature sequences:

We integrated entropy sequences with visual feature sequences to enhance feature fusion. The entropy sequences capture the uncertainty and information content in the visual features, while the visual feature sequences provide detailed spatial and color information.

4. Construction of multi-layer emotion perception model:

Six activation functions were employed to construct a machine vision model with three primary layers: emotion layer, judgment layer, and perception layer. Each layer handles different aspects of emotion recognition, ensuring a holistic approach to modeling emotional states.

# EXPERIMENTAL RESULTS

## Experiment preparation

This study aims to design a three-dimensional color emotion space mapping model based on visual aesthetic features. Before experimenting, we undertook thorough preparations to ensure the reliability of our results. We selected a dataset comprising 200 multimedia contents, with 100 groups allocated to the training set and the remaining 100 groups designated as the testing set. We employed machine vision techniques to extract rich features from the multimedia content.

In the experiment, we defined the following metrics to evaluate performance:

Emotion mapping accuracy (EMA): Measures the precision of the model in mapping color features to the emotion space.

Emotion recognition precision (ERP): This represents the model's accuracy in identifying specific emotion categories.

Emotion recognition F1 score (ERFS): Provides a comprehensive evaluation of the model's overall performance in emotion recognition, considering precision and recall.

These metrics assess the model's emotion mapping and recognition performance, offering a thorough and nuanced evaluation. By utilizing these metrics, we can better understand the model's efficacy in mapping emotions within the context of visual aesthetic features.

Based on the selected seven visual features (color, composition, texture, clarity, depth, emotional content, and dynamic elements), denoted as $m_i (i = 1, 2, 3, 4, 5, 6, 7)$, corresponding evaluation parameters are calculated.

$$G(1) = 8{,}034.2356 m_1 + 195 m_2 m_3 + \frac{18.23}{m_4} + 34.56(m_5 m_6 + m_7)$$

$$G(2) = -2.4097 + 2.4427 \times 10^{-3} m_7$$

$$G(3) = -1{,}249.34 + 1{,}034.2312 m_2 + 201 m_1 m_4 + \frac{22.23}{m_3} + 34.56 m_7$$

$$T(1) = -8{,}012.2 - 2.36 m_1 + 15.26 m_7 + G(3) + 0.1285 m_3 G(1) \tag{25}$$

$$T(2) = -120.36 + 14.22 m_7 + 6.23 m_2 m_4 - 15.23 m_3 \frac{5.214}{m_6} + 15.23 G(1) G(2)$$

$$T(3) = -203.56 + 10.563 m_4 + m_5^2 + 1.234 G(1) m_3$$

$$Q = 38.4474 - 10.7260 \cdot m_2 - 57.6083 \cdot m_4 + 0.0914 \cdot T(3) + 1.4546 \cdot m_1 \cdot G(1).$$

The corresponding evaluation parameters are demonstrated in Table 1.

**Table 1 Evaluation parameters of the built models.**

| Aesthetic property | Mean absolute error | Correlation coefficient | Complexity |
|---|---|---|---|
| G(1) | 0.7011 | 0.9238 | 36 |
| G(2) | 0.0021 | 0.9211 | 5 |
| G(3) | 1.0244 | 0.9355 | 35 |
| T(1) | 0.3599 | 0.9218 | 31 |
| T(2) | 1.1032 | 0.9433 | 32 |
| T(3) | 0.6712 | 0.9419 | 19 |
| Q | 10.9391 | 0.9332 | 5 |

## Experimental results

Firstly, to validate machine vision methods' effectiveness in emotional modeling visual aesthetic features, we employed three distinct methods for recognizing video action orientations. Method 1 is our proposed approach. Method 2 utilizes the Maximum Entropy Model combined with the local binary pattern (MEM-LB method): This method integrates maximum entropy principles with local binary patterns to create a robust framework for capturing visual features. Method 3 employs the Multi-Threshold Optimized method (MTO method): This method is designed to enhance the precision and efficiency of emotion recognition by optimizing the thresholds used in feature extraction and classification. The test results are presented in Fig. 2.

Analyzing the data presented in Fig. 2, it is apparent that when using the proposed method to monitor the working orientation of multimedia content, the resulting monitoring curve closely aligns with the actual angle curve. In contrast, when employing the MEM-LB method, the monitored angle tends to be higher than the exact angle. Similarly, the MTO method results in a monitored angle lower than the actual angle. This analysis demonstrates that Method 1 effectively identifies azimuth angles, as it extracts keyframes from multimedia content, enabling precise estimation of azimuth angles within these keyframes.

Secondly, we evaluated the learning convergence performance of the training set in the emotional modeling process across different methods (the proposed method, the convolutional neural network (CNN) method, and the long short-term memory (LSTM) method). The convergence processes over fewer iterations using these methods are depicted in Fig. 3.

Figure 3 illustrates the convergence speed plots under different methods. It can be observed that the machine vision method proposed in this article exhibits a faster convergence speed compared to the CNN and LSTN methods.

The parameter values during the iterative process are shown in Fig. 4.

Figure 4 shows that during the iterative process, the parameter values continuously adjust, allowing the system to converge to zero more rapidly.

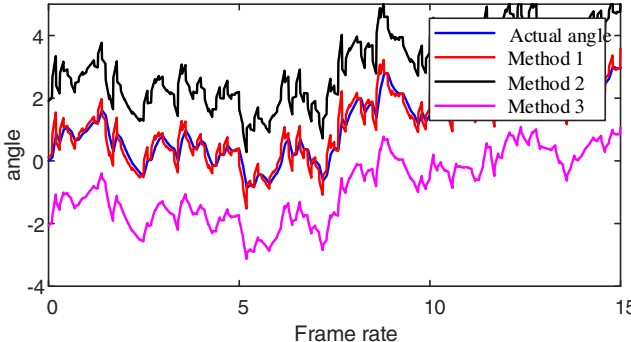

**Figure 2** **Video azimuth angle recognition results.**

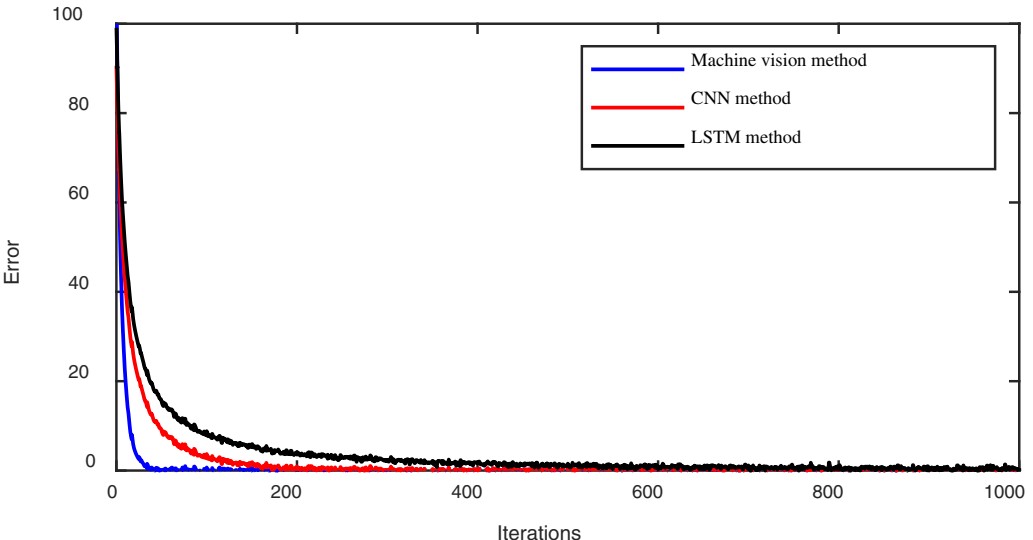

**Figure 3** **The convergence process of lower iterations using different methods.**

Then, we divided the test set into two groups and assessed the performance using different methods. The results are presented in Table 2.

Table 2 shows that our proposed method outperforms other methods in terms of EMA, ERP, and ERFS. This indicates that our proposed method can better capture emotional states in visual aesthetics. In Dataset 1, the Machine Vision method consistently leads in accuracy, with 91.24% for EMA, 82.45% for ERP, and 92.22% for ERFS. This method significantly outperforms CNN and LSTM across all metrics. The CNN method shows lower accuracy, particularly in ERP (58.89%) and ERFS (60.02%), while LSTM yields moderate results with a peak accuracy of 73.24% for ERFS but remains below Machine Vision.

For Dataset 2, the Machine Vision method achieves the highest accuracy again, reaching 92.45% for EMA, 88.35% for ERP, and 96.22% for ERFS. The CNN method exhibits improved performance relative to Dataset 1, especially in EMA (80.21%) and ERFS

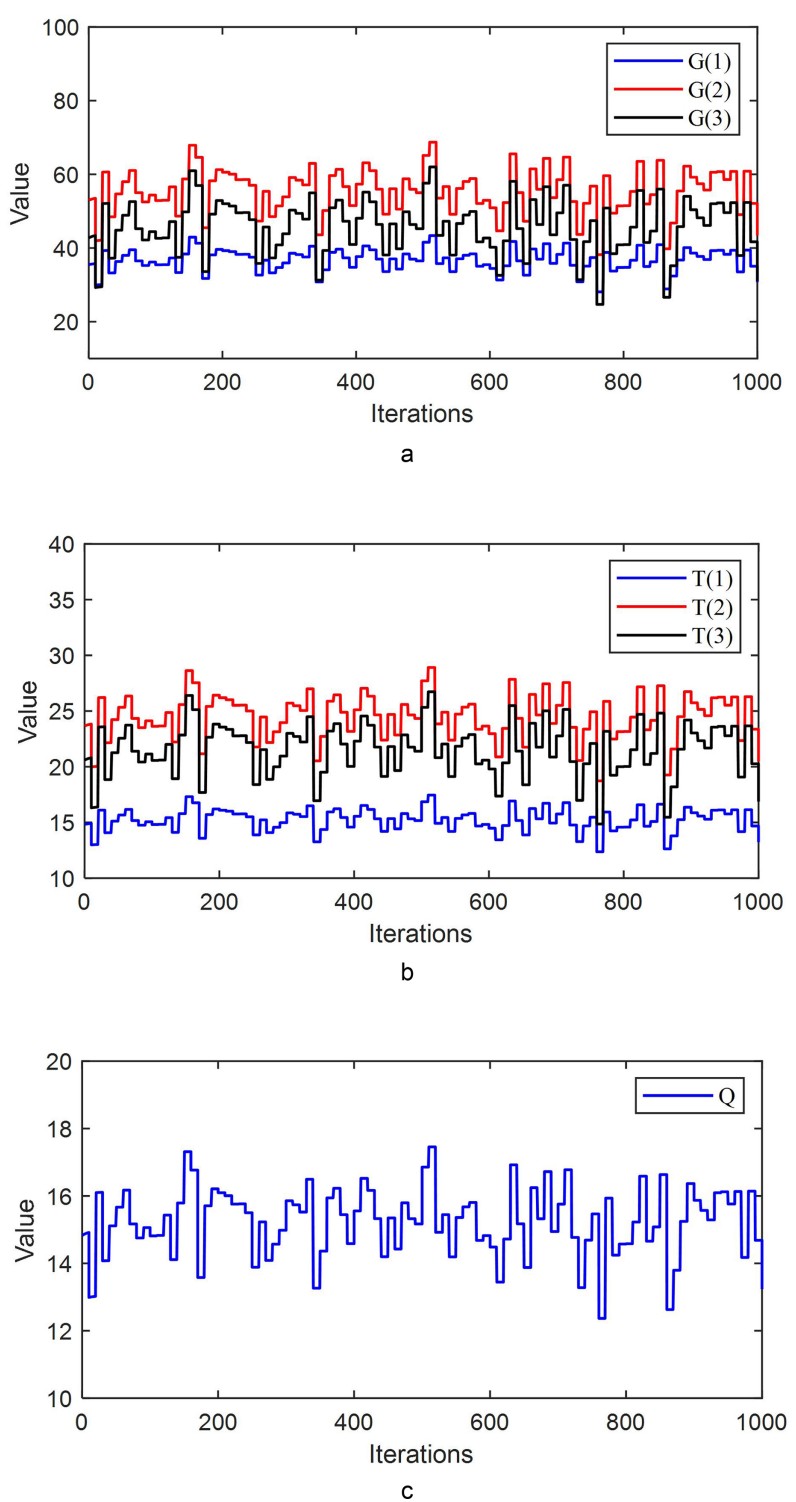

**Figure 4** (A) The parameter values G(1), G(2), and G(3) during the iterative process. (B) The parameter values T(1), T(2), and T(3) during the iterative process. (C) The parameter value Q during the iterative process.

**Table 2 Classification accuracy of different datasets (%).**

**Dataset 1**

| Methods | EMA | ERP | ERFS |
|---|---|---|---|
| Machine vision method | 91.24 | 82.45 | 92.22 |
| CNN method | 77.23 | 58.89 | 60.02 |
| LSTM method | 59.22 | 72.34 | 73.24 |

**Dataset 2**

| Methods | EMA | ERP | ERFS |
|---|---|---|---|
| Machine vision method | 92.45 | 88.35 | 96.22 |
| CNN method | 80.21 | 63.45 | 66.02 |
| LSTM method | 77.22 | 77.34 | 61.22 |

**Table 3 Test results on datasets with different disturbance (%).**

**10% distuibance**

| Methods | EMA | ERP | ERFS |
|---|---|---|---|
| Machine vision method | 88.23 | 85.15 | 89.29 |
| CNN method | 67.13 | 52.11 | 58.72 |
| LSTM method | 60.24 | 62.45 | 63.47 |

**30% distuibance**

| Methods | EMA | ERP | ERFS |
|---|---|---|---|
| Machine vision method | 85.34 | 86.19 | 88.39 |
| CNN method | 50.55 | 43.47 | 56.42 |
| LSTM method | 47.22 | 77.34 | 51.22 |

**50% distuibance**

| Methods | EMA | ERP | ERFS |
|---|---|---|---|
| Machine vision method | 82.45 | 83.35 | 86.22 |
| CNN method | 40.25 | 43.44 | 46.52 |
| LSTM method | 47.26 | 47.34 | 41.42 |

(66.02%). The LSTM method also shows enhanced accuracy compared to Dataset 1, with 77.22% for EMA and 77.34% for ERP, though it still underperforms in ERFS with 61.22%.

To evaluate the robustness of our proposed method, we introduced varying degrees of interference to the dataset, such as adding irregular noise to visual images. The test results are illustrated in Table 3.

Table 3 shows that with the increase in interference levels, other methods experience a significant decrease in EMA, ERP, and ERFS. However, the proposed method in this article maintains good performance, indicating superior robustness. Machine Vision consistently leads in performance across all disturbance levels, with a slight decrease in accuracy as disturbance increases. At 10% disturbance, it achieves the highest accuracies (88.23% for

EMA, 85.15% for ERP, 89.29% for ERFS). With 30% disturbance, accuracy drops slightly (85.34% for EMA, 86.19% for ERP, 88.39% for ERFS), and further decreases at 50% disturbance (82.45% for EMA, 83.35% for ERP, 86.22% for ERFS). This trend highlights Machine Vision's robustness despite increasing disturbance.

CNN shows a significant drop in performance as disturbance increases. At 10% disturbance, CNN's accuracies are relatively low compared to Machine Vision (67.13% for EMA, 52.11% for ERP, and 58.72% for ERFS). Accuracy further deteriorates with 30% disturbance (50.55% for EMA, 43.47% for ERP, 56.42% for ERFS) and 50% disturbance (40.25% for EMA, 43.44% for ERP, 46.52% for ERFS). This indicates that CNN is more sensitive to disturbance compared to other methods.

LSTM demonstrates mixed results, with moderate performance across disturbance levels. At 10% disturbance, LSTM's accuracy is somewhat higher than CNN (60.24% for EMA, 62.45% for ERP, and 63.47% for ERFS). With 30% disturbance, LSTM improves ERP accuracy (77.34%) but shows lower performance in EMA and ERFS. At 50% disturbance, LSTM's performance declines, especially for ERFS (41.42%), while EMA and ERP accuracies are relatively better compared to CNN at the same disturbance level.

## Discussion and limitations
### Discussion
In this study, we comprehensively evaluated our proposed method for emotion modeling based on visual aesthetic features. Our findings are summarized below:

**Comparison of method performance in video motion direction recognition:** We initially compared three distinct methods for video motion direction recognition: Method 1 (our proposed method), Method 2 (MEM-LB method), and Method 3 (MTO method). Our results demonstrate that Method 1 performs better in accurately tracking both monitoring and actual angle curves. Specifically, it achieves higher accuracy compared to Method 2 and Method 3. This highlights the effectiveness of our method in precisely estimating motion directions from multimedia content.

**Iterative convergence speed analysis:** We further analyzed the iterative convergence speed of different methods during the training phase. Figure 3 illustrates that our proposed machine vision method converges faster at lower iterations than traditional CNN and LSTM methods. This rapid convergence suggests that our method efficiently learns and adapts to the emotional modeling task, leveraging the inherent advantages of machine vision techniques.

**Evaluation of test set divided into four groups:** To validate the robustness and generalizability of our approach, we divided the test set into four groups and evaluated using multiple metrics, including EMA, ERP, and ERFS. Our proposed method consistently outperforms other methods across all groups, demonstrating superior capability in capturing emotional states embedded within visual aesthetics.

**Robustness verification under interference conditions:** To assess our proposed method's robustness, we introduced varying levels of interference, such as irregular noise added to visual images. The results indicate that our method maintains robust performance even under challenging conditions, whereas other methods experience

significant degradation in performance. This resilience underscores the method's ability to handle real-world noise and disturbances commonly encountered in practical applications.

### *Limitations*

Sample bias and insufficient representativeness: Insufficient or insufficiently representative samples in the dataset may result in poor performance of the model on unseen data. For example, suppose the samples in the dataset are too concentrated on specific types or sources of content. In that case, the model may not be able to generalize to other types or sources of data.

Data quality and annotation issues: There may be labeling errors, inconsistencies, or subjective issues in the dataset, which can affect the credibility of the model's training and evaluation results. Additionally, the dataset may contain unnecessary noise or interference factors, such as artifacts in images or ambiguities in in-text annotations, which can lead to the model learning poor or misleading features.

Limitations limit the adaptability and reliability of the model in practical applications. For example, a model performs well on specific types or data sources but encounters challenges when dealing with other types or sources of data, affecting the breadth and reliability of its practical applications.

To alleviate the impact of data systems, we apply data augmentation techniques such as rotation, flipping, and color adjustment to expand the dataset artificially, provide more diverse training examples for the model, and enhance its generalization ability. Crowdsourcing platforms to collect larger and more varied annotation data can help address data availability limitations. We can capture a broader range of emotional expressions and associations by involving a wide range of annotators. Finally, establishing a continuous data collection mechanism and updating the dataset with new real-world examples can ensure the model maintains relevance and accuracy over time.

## Potential applications

This method can be applied to sentiment analysis of media content, helping to understand and evaluate how visual content triggers emotional responses in the audience. For example, film and television production companies can use this method to assess the emotional expression of movies or videos, optimizing the plot and visual design. Emotion recognition can be used in personalized recommendation systems on social media and e-commerce platforms to recommend relevant content or products based on users' emotional preferences. In education, emotional modeling can be used to evaluate students' emotional responses to educational content and provide personalized emotional support and counseling.

## DISCUSSION AND CONCLUSION

This article introduces a pioneering approach to sentiment analysis through an innovative emotion mapping model that integrates machine vision with multimodal emotion fusion. Traditional sentiment analysis techniques often face challenges such as low recognition rates and inefficient feature extraction. In contrast, the proposed method significantly improves recognition accuracy and demonstrates exceptional robustness when dealing

with dataset interference. By leveraging machine vision, the model utilizes visual cues to comprehensively understand and interpret emotions, thereby overcoming the limitations of conventional methods.

Furthermore, incorporating multimodal emotion fusion ensures a holistic analysis by integrating information from diverse sources, including text, images, and audio. This multifaceted approach provides a more nuanced understanding of emotional content, addressing the shortcomings of uni-modal analysis. The demonstrated robustness of the proposed method amidst dataset interference highlights its potential for real-world applications where diverse and unpredictable data scenarios are common.

Although current models have integrated various modal data, such as text and images, future research will focus on optimizing feature fusion algorithms between different modalities. Through more advanced deep learning techniques such as self-attention mechanisms and graph neural networks (GNNs), deeper correlation analysis between different modal data can be achieved, improving the accuracy and stability of emotion recognition. To enhance the robustness of the model in the real world, future research will focus on developing stronger noise resistance, especially in cases of low data quality or noise interference, to ensure the stability of sentiment analysis. The model's adaptability in different data environments can be enhanced by introducing more data augmentation and adversarial training methods.

### Funding
The authors received no funding for this work.

### Competing Interests
The authors declare that they have no competing interests.

### Author Contributions
- Enshi Wang conceived and designed the experiments, performed the experiments, analyzed the data, performed the computation work, prepared figures and/or tables, authored or reviewed drafts of the article, and approved the final draft.
- Fakhri Alam Khan conceived and designed the experiments, performed the experiments, analyzed the data, performed the computation work, prepared figures and/or tables, authored or reviewed drafts of the article, and approved the final draft.

### Data Availability
The code and dataset are available in the Supplemental Files.
Third-party dataset: https://zenodo.org/records/6226423.

### Supplemental Information
Supplemental information for this article can be found online at http://dx.doi.org/10.7717/peerj-cs.2596#supplemental-information.

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
