# Peer review of "Design of a 3D emotion mapping model for visual feature analysis using improved Gaussian mixture models"

_PeerJ Computer Science, doi:10.7717/peerj-cs.2596_

## Round 0.1 · original submission · Major Revisions

The reviewers have commented on the manuscript, which is a resubmission of your previously rejected manuscript, and their views are broadly positive. Please revise the paper addressing their comments.

However, there were some concerns that parts of your manuscript may have been written with the aid of AI tools. We would like to remind you that according to our policy on this respect, which you can find at https://peerj.com/about/policies-and-procedures/#publication-ethics. If you did make use of any such tool, then you should declare this fact. Alternatively, you can remove the AI-generated text, if any.

Reviewer 1 ·

Basic reporting

Overall, the authors took the Reviewers’ suggestions into account and the paper is much improved.
However, I still think that this paper's content does not refer to its title. This concerns ‘auditory modality.‘ It did not fully capture the paper’s scope if audio data were not explicitly analyzed and integrated.
The current manuscript predominantly focuses on visual features, particularly color-emotion space mapping, while the title claims multimodal data integration.

Therefore, I suggest changing the title of this paper to:
"Design of a 3D Emotion Mapping Model for Visual Feature Analysis Using Improved Gaussian Mixture Models"

or
“3D Color-Emotion Space Model for Enhanced Visual Feature Extraction Using Improved Gaussian Mixture Models"

or

“3D Color-Emotion Feature Extraction Framework for Visual Data Integration"

Moreover, in the conclusion or future work sections (and potential applications), the authors may outline plans to expand the model for more complex audio-visual interactions or address specific scenarios where both modalities significantly improve emotional understanding.

Also, below, there are examples that need the authors’ attention.

denote Euclidian norm -> denote the Euclidian norm
The findings indicate that significantly enhances emotional accuracy. -> this sentence is not comprehensible
metrics including -> metrics, including
insufficie2nt
machine learning method s-> machine-learning

Experimental design

As mentioned above, the experimental design is much clearer.

Validity of the findings

As above - their description was improved.

Additional comments

Be more careful when correcting your paper.

Reviewer 2 ·

Basic reporting

1. I would like to thank the authors for their thorough revisions and improvements to the abstract. The revision partially addresses the concerns related to the abstract. However, the concern "the abstract lacks a clear explanation of how this model fits into the broader field and why it's important in comparison to traditional methods" was not reflected in the updated paper.

2. As previously recommended, it’s essential to have a fluent English speaker or a professional editing service go through the entire paper, not the abstract only. While the readability has improved, there are still occasional grammatical errors and formatting inconsistencies. For example, "Based on the selected 7 visual features", it should be "seven" instead of "7". Some prepositions feel awkward like "This method exhibits strong performance with a high Emotion Mapping Accuracy" could be improved to "This method exhibits strong performance in achieving high Emotion Mapping Accuracy."

3. The concern regarding the referencing of recent studies is resolved. The manuscript now includes sufficient citations from the past two to three years.

4. Some figures still lack readable, clear labels, please, increase the font.

5. Please, support the statements you make throughout the paper, especially in Related Work. For example, "Traditional sentiment analysis methods are often too simplistic to accurately capture the nuanced emotions present in multimedia content. " Please, cite the references to support your claims.

Experimental design

The paper now includes tables that provide detailed results from experimental evaluations.

The revised paper provides more information at the beginning of the Methodology section, a better introduction to the proposed approach before transitioning into its components​, but still, it is somewhat general. I would recommend making it more specific, just mention all steps/methods you employ and describe in this section, e.g., Keyframe extraction based on machine vision, Working sequence potential function, etc., to see the whole picture and then jump to details.

Validity of the findings

The previous concern related to the Discussion was partially addressed. However, the discussion of the results is still somewhat limited. Discussion is not only about restating results. You can include, for example:
-Comparing the results with recent studies and common findings.
- Reflecting on common methodological challenges and how the current work addresses or shares these challenges.

Additional comments

-

---

## Round 0.2 · accepted · Accept

The reviewers are overall happy with the current revision, and I am pleased to accept this manuscript for publication in PeerJ Computer Science.

Reviewer 1 ·

Basic reporting

Overall, I am satisfied with the revised paper version. Especially important were changes to the tile and Abstract.

Experimental design

This is sufficiently comprehensible as is.

Validity of the findings

As above, the tabular presentation of the results is clear.

Additional comments

The quality of the figures could still be improved. However, I suppose, this can be done during the paper editing stage.